# On importance-weighted autoencoders

## Abstract

The *importance weighted autoencoder (IWAE)* (Burda et al., 2016) is a popular variational-inference method which achieves a tighter evidence bound (and hence a lower bias) than standard variational autoencoders by optimising a *multi-sample objective,* i.e. an objective that is expressible as an integral over $K > 1$ Monte Carlo samples. Unfortunately, the IWAE multi-sample objective leads to inference-network gradients which break down as $K$ is increases (Rainforth et al., 2018). This breakdown can only be circumvented by removing high-variance score-function terms, either by heuristically ignoring them (which yields the *'sticking-the-landing' IWAE (IWAE-STL)* gradient from Roeder et al. (2017)) or through an identity from Tucker et al. (2019) (which yields the *'doubly-reparametrised' IWAE (IWAE-DREG)* gradient). In this work, we develop an encompassing framework which directly optimises the proposal distribution in importance sampling as in the *reweighted wake-sleep (RWS)* algorithm from Bornschein & Bengio (2015). From this unified framework, most of the previously proposed gradient estimators can be naturally derived. This permits a better understanding of the assumptions and trade-offs that are at play. Importantly, the derived gradient estimators are guaranteed to not degenerate as $K \to \infty$.

## 1 Introduction

Let $x$ be some observation and let $z$ be some latent variable taking values in some space $\mathsf{Z}$. These are modeled via the *generative model* $p_\theta(z, x) = p_\theta(z)p_\theta(x|z)$ which gives rise to the marginal likelihood $p_\theta(x) = \int_{\mathsf{Z}} p_\theta(z, x) \, \mathrm{d}z$ of the model parameters $\theta$. In this work, we analyse algorithms for *variational inference,* i.e. algorithms which aim to

1. learn the generative model, i.e. find a value $\theta^\star$ which is approximately equal to the *maximum-likelihood estimate (MLE)* $\theta^{\mathrm{ML}} \coloneqq \arg\max_\theta p_\theta(x)$;

2. construct a tractable *variational approximation* $q_{\phi,x}(z)$ of $p_\theta(z|x) = p_\theta(z, x)/p_\theta(x)$, i.e. find the value $\phi^\star$ such that $q_{\phi^\star,x}(z)$ is as close as possible to $p_\theta(z|x)$ in some suitable sense.

A few comments about this setting are in order. Firstly, as is common in the literature, we restrict our presentation to a single latent representation–observation pair $(z, x)$ to avoid notational clutter – the extension to multiple independent observations is straightforward. Secondly, we assume that no parameters are shared between the generative model $p_\theta(z, x)$ and the variational approximation $q_{\phi,x}(z)$. This is common in neural-network applications but could be relaxed. Thirdly, our setting is general enough to cover amortised inference. For this reason, we often refer to $\phi$ as the parameters of an *inference network.*

Two main classes of stochastic gradient-ascent algorithms for optimising $\psi \coloneqq (\theta, \phi)$ which employ $K \geq 1$ Monte Carlo samples ('particles') to reduce errors have been proposed.

- **IWAE.** The *importance weighted autoencoder (IWAE)* (Burda et al., 2016) maximizes a joint lower bound $\mathcal{L}_\psi^K \leq p_\theta(x)$ whose bias decreases as $K \to \infty$. The gradients of this objective can be unbiasedly approximated via the Monte-Carlo method. Unfortunately, the signal-to-noise ratio of the IWAE $\phi$-gradient vanishes as $K$ grows (Rainforth et al., 2018). Two modified IWAE $\phi$-gradients avoid this breakdown by removing high-variance 'score-function' terms:

- **IWAE-STL.** The *'sticking-the-landing' IWAE (IWAE-STL)* $\phi$-gradient (Roeder et al., 2017) heuristically drops the problematic score-function terms from the IWAE $\phi$-gradient. This induces bias for the IWAE objective.
- **IWAE-DREG.** The *'doubly-reparametrised' IWAE (IWAE-DREG)* $\phi$-gradient (Tucker et al., 2019) unbiasedly removes the problematic score-function terms from the IWAE $\phi$-gradient using a formal identity.

- **RWS.** The *reweighted wake-sleep (RWS)* algorithm (Bornschein & Bengio, 2015) optimises two separate objectives for $\theta$ and $\phi$. Its gradients are approximated by self-normalised importance sampling with $K$ particles: this induces a bias which vanishes as $K \to \infty$. RWS can be viewed as an adaptive importance-sampling approach which iteratively improves its proposal distribution while simultaneously optimising $\theta$ via stochastic approximation. Crucially, the RWS $\phi$-gradients do not degenerate as $K \to \infty$.

Of these two methods, the IWAE is the most popular and Tucker et al. (2019) demonstrated empirically that RWS can break down, conjecturing that this is due to the fact that RWS does not optimise a joint objective (for $\theta$ and $\phi$). Meanwhile, the IWAE-STL gradient performed consistently well despite lacking a firm theoretical footing. Yet, IWAE suffers from the above-mentioned $\phi$-gradient breakdown and exhibited inferior empirical performance to RWS (Le et al., 2019). Thus, it is not clear whether the multi-sample objective approach of IWAE or the adaptive importance-sampling approach of RWS is preferable.

In this work, we show that directly optimising the proposal distribution, e.g. as done by RWS, is preferable to optimising the IWAE multi-sample objective because (a) the multi-sample objective typically relies on reparametrisations and, even if these are available, leads to the $\phi$-gradient breakdown, (b) modifications of the IWAE $\phi$-gradient which avoid this breakdown (i.e. IWAE-STL and IWAE-DREG) can be justified in a more principled manner by taking an RWS-type adaptive importance-sampling view. This conclusion was already reached by Le et al. (2019) based on numerical experiments. They demonstrated that the need for reparametrisations can make IWAE inferior to RWS e.g. for discrete latent variables. Our work complements theirs by formalising this argument. To this end, we slightly generalise the RWS algorithm to obtain a generic adaptive importance-sampling framework for variational inference which we term *adaptive importance sampling for learning (AISLE)* for ease of reference. We then show that AISLE admits not only RWS but also the IWAE-DREG and IWAE-STL gradients as special cases.

**Contributions.** Novel material is presented in Section 3, where we introduce the AISLE-framework. From this, most of the previously proposed gradient estimators can be naturally derived in a principled manner. Importantly, the derived gradient estimators are guaranteed to not degenerate as $K \to \infty$. Specifically, we establish the following connections.

- We prove that the IWAE-STL gradient can be recovered as a special case of AISLE via a principled and novel application of the 'double-reparametrisation' identity from Tucker et al. (2019). This indicates that the breakdown of RWS observed in Tucker et al. (2019) may not be due to its lack of a joint objective as previously conjectured (since IWAE-STL avoided this breakdown despite having the same idealised objective as RWS). Our work also provides a theoretical foundation for IWAE-STL which was hitherto only heuristically justified as a biased IWAE-gradient.

- We prove that AISLE also admits the IWAE-DREG gradient as a special case. Our derivation also makes it clear that the learning rate should be scaled as $\mathcal{O}(K)$ for the IWAE $\phi$-gradient (and its modified version IWAE-DREG) unless the gradients are normalised as implicitly done by popular optimisers such as ADAM (Kingma & Ba, 2015). In contrast, the learning rate for AISLE need not be scaled up with of $K$.

- When applied to the family of $\alpha$-divergences, AISLE leads to a new family of gradient estimators that generalises some previously derived in the literature.

- In the supplementary materials, we provide insights into the impact of the self-normalisation bias on some of the importance-sampling based gradient approxima-

tions (Appendix A) and empirically compare the main algorithms discussed in this work (Appendix B).

We stress that the focus of our work is not necessarily to derive new algorithms nor to establish which of the various special cases of AISLE is preferable. Indeed, while we compare all algorithms discussed in this work empirically on Gaussian models in the supplementary materials, we refer the reader to Tucker et al. (2019); Le et al. (2019) for an extensive empirical comparisons of all the algorithms discussed in this work.

**Notation.** We repeatedly employ the shorthand $p(f) := \int_{\mathsf{Z}} f(z)p(z)\,\mathrm{d}z$ for the integral of some $p$-integrable test function $f$; thus, $p(f) = \mathbb{E}_{z \sim p}[f(z)]$ if $p$ is a probability measure. Furthermore, $q^{\otimes K}(z^{1:K}) := \prod_{k=1}^{K} q(z^k)$. To keep the notation concise, we hereafter suppress dependence on the observation $x$, i.e. we write $q_\phi(z) := q_{\phi,x}(z)$ as well as

$$\pi_\theta(z) := p_\theta(z|x) = \frac{p_\theta(z,x)}{p_\theta(x)} = \frac{\gamma_\theta(z)}{\mathcal{Z}_\theta},$$

where $\gamma_\theta(z) := p_\theta(z,x)$ and where $\mathcal{Z}_\theta := p_\theta(x) = \int_{\mathsf{Z}} \gamma_\theta(z)\,\mathrm{d}z$.

## 2 BACKGROUND

### 2.1 IMPORTANCE SAMPLING

The expectation $q_\phi(f)$ of a test function $f : \mathsf{Z} \to \mathbb{R}$ can be unbiasedly estimated by the quantity $[f(z^1) + \ldots + f(z^K)]/K$ using a set of $K$ particles, $\mathbf{z} := (z^1, \ldots, z^K) \sim q_\phi^{\otimes K}$, which are independent and identically distributed (IID) according to $q_\phi$. Similarly, expectations of the type $\pi_\theta(f)$ can be approximated by the *self-normalised importance sampling* estimate

$$\hat{\pi}_\theta\langle\phi, \mathbf{z}\rangle(f) := \sum_{k=1}^{K} \bar{w}_\psi^k\, f(z^k) \quad \text{with} \quad \bar{w}_\psi^k := \frac{w_\psi(z^k)}{\sum_{l=1}^{K} w_\psi(z^l)} \quad \text{and} \quad w_\psi(z) := \frac{\gamma_\theta(z)}{q_\phi(z)}.$$

The notation $\langle\phi, \mathbf{z}\rangle$ stresses the dependence of this estimator on $\phi$ and $\mathbf{z}$. The quantity $w_\psi(z^k)$ are called the $k$th importance weight and $\bar{w}_\psi^k$ is its self-normalised version. For readability, we have dropped the dependence of $\bar{w}_\psi^k$ on $\mathbf{z} \in \mathsf{Z}^K$ from the notation.

**Remark 1.** *Since $\mathcal{Z}_\theta = q_\theta(w_\psi)$, the quantity $\widehat{\mathcal{Z}}_\theta\langle\phi, \mathbf{z}\rangle := [w_\psi(z^1) + \ldots + w_\psi(z^K)]/K$ is an unbiased ('importance-sampling') estimator of $\mathcal{Z}_\theta$.*

**Remark 2.** *The self-normalised estimate $\hat{\pi}_\theta\langle\phi, \mathbf{z}\rangle(f)$ is typically not unbiased. Under mild assumptions (e.g. if $\sup w_\psi < \infty$), its bias vanishes at rate $\mathcal{O}(K^{-1})$, its standard deviation vanishes at Monte-Carlo rate $\mathcal{O}(K^{-1/2})$ and $\hat{\pi}_\theta\langle\phi, \mathbf{z}\rangle(f) \to \pi_\theta(f)$ almost surely as $K \to \infty$.*

### 2.2 IMPORTANCE WEIGHTED AUTOENCODER (IWAE)

**Objective.** The *importance weighted autoencoder (IWAE)*, introduced by Burda et al. (2016), seeks to find a value $\theta^\star$ of the generative-model parameters $\theta$ which maximises a lower bound $\mathcal{L}_\psi^K$ on the log-marginal likelihood ('evidence'). This bound depends on the inference-network parameters $\phi$ and the number of samples, $K \geq 1$:

$$\psi^\star := (\theta^\star, \phi^\star) := \arg\max_\psi \mathcal{L}_\psi^K, \quad \mathcal{L}_\psi^K := \mathbb{E}\big[\log \widehat{\mathcal{Z}}_\theta\langle\phi, \mathbf{z}\rangle\big]. \tag{1}$$

where the expectation is w.r.t. $\mathbf{z} \sim q_\phi^{\otimes K}$. For any finite $K$, optimisation of the inference-network parameters $\phi$ tightens the evidence bound. Burda et al. (2016) prove that for any $\phi$ we have that $\mathcal{L}_\psi^K \uparrow \log \mathcal{Z}_\theta$ as $K \to \infty$. If $K = 1$, the IWAE reduces to the variational autoencoder (VAE) from Kingma & Welling (2014). However, for $K > 1$, as pointed out in Cremer et al. (2017); Domke & Sheldon (2018), the IWAE also constitutes another VAE on an extended space based on an auxiliary-variable construction developed in Andrieu & Roberts (2009); Andrieu et al. (2010); Lee (2011) (see, e.g. Finke, 2015, for a review).

**Standard reparametrisation gradient.** The gradient of the IWAE objective from (1):
$\nabla_\psi \mathcal{L}_\psi^K = \mathbb{E}\big[\nabla_\psi \log \widehat{\mathcal{Z}}_\theta \langle \phi, \mathbf{z} \rangle + G_\psi(\mathbf{z})\big]$, with $G_\psi(\mathbf{z}) := \log \widehat{\mathcal{Z}}_\theta \langle \phi, \mathbf{z} \rangle \sum_{k=1}^K \nabla_\psi \log q_\phi(z^k)$. The
intractable quantity $\mathbb{E}\big[G_\psi(\mathbf{z})\big]$ can be approximated unbiasedly via a vanilla Monte Carlo
approach using a single ($K$-dimensional) sample point $\mathbf{z} = (z^1, \ldots, z^K) \sim q_\phi^{\otimes K}$. Unfortunately,
this approximation typically has such a large variance that it becomes impracticably noisy
(Paisley et al., 2012). To remove this high-variance term, the well known *reparametrisation
trick* (Kingma & Welling, 2014) is usually employed. It requires the following assumption.

(**R1**) *There exists a distribution $q_\epsilon$ on some space $\mathsf{E}$ and a family of differentiable mappings
$h_\phi \colon \mathsf{E} \to \mathsf{Z}$ such that if $\epsilon \sim q_\epsilon$ we have that $z = z(\epsilon) = h_\phi(\epsilon) \sim q_\phi$.*

Under **R1**, with $\epsilon^1, \ldots, \epsilon^K \overset{\text{IID}}{\sim} q_\epsilon$ and $z^k := z(\epsilon^k) := h_\phi(\epsilon^k)$, the gradient can be expressed as

$$\nabla_\psi \mathcal{L}_\psi^K = \mathbb{E}\left[\sum_{k=1}^K \bar{w}_\psi^k \, \nabla_\psi \log w_\psi(z^k)\right] = \mathbb{E}\left[\sum_{k=1}^K \bar{w}_\psi^k \, \begin{pmatrix} \nabla_\theta \log \gamma_\theta(z^k) \\ \blacktriangledown_\psi(z^k) - \nabla_\phi \log q_\phi(z^k) \end{pmatrix}\right], \qquad (2)$$

with $\blacktriangledown_\psi(z) := \nabla_\phi[\log \circ \, w_{\psi'} \circ h_\phi]|_{\psi'=\psi}(h_\phi^{-1}(z))$. Here, the notation $\psi'$ indicates that one does
not differentiate $w_\psi$ w.r.t. $\psi$. The IWAE then uses a vanilla Monte Carlo estimate of (2),

$$\begin{bmatrix} \widehat{\nabla}_\theta^{\text{IWAE}}\langle \phi, \mathbf{z} \rangle \\ \widehat{\nabla}_\phi^{\text{IWAE}}\langle \theta, \mathbf{z} \rangle \end{bmatrix} := \sum_{k=1}^K \bar{w}_\psi^k \begin{bmatrix} \nabla_\theta \log \gamma_\theta(z^k) \\ \blacktriangledown_\psi(z^k) - \nabla_\phi \log q_\phi(z^k) \end{bmatrix}. \qquad (3)$$

Before proceeding, we state the following lemma, proved in Tucker et al. (2019, Section 8.1),
which generalises of the well-known identity $q_\phi(\nabla_\phi \log q_\phi) = 0$.

**Lemma 1 (Tucker et al. (2019)).** *Under **R1**, for suitably integrable $f_\psi \colon \mathsf{Z} \to \mathbb{R}$, we have*

$$q_\phi(f_\psi \nabla_\phi \log q_\phi) = q_\epsilon(\nabla_\phi[f_{\psi'} \circ h_\phi]|_{\psi'=\psi}) = q_\phi(\nabla_\phi[f_{\psi'} \circ h_\phi]|_{\psi'=\psi} \circ h_\phi^{-1}).$$

We now exclusively focus on the $\phi$-portion of the IWAE gradient, $\widehat{\nabla}_\phi^{\text{IWAE}}\langle \theta, \mathbf{z} \rangle$.

**Remark 3 (drawbacks of the IWAE $\phi$-gradient).** *The gradient $\widehat{\nabla}_\phi^{\text{IWAE}}\langle \theta, \mathbf{z} \rangle$ has three
drawbacks. The last two of these are attributable to the 'score-function' terms $\nabla_\phi \log q_\phi(z)$
in the $\phi$-gradient portion of (3).*

- **Reliance on reparametrisations.** *A reparametrisation à la **R1** is necessary to
  remove the high-variance term $G_\psi(\mathbf{z})$. For, e.g. discrete, models that violate **R1**,
  control-variate approaches (Mnih & Rezende, 2016) or continuous relaxations have
  been proposed but these incur additional implementation, tuning and computation
  costs whilst not necessarily reducing the variance (Le et al., 2019).*

- **Vanishing signal-to-noise ratio.** *The $\phi$-gradient breaks down in the sense that
  its signal-to-noise ratio vanishes as $\mathbb{E}[\widehat{\nabla}_\phi^{\text{IWAE}}\langle \theta, \mathbf{z} \rangle]/\operatorname{var}[\widehat{\nabla}_\phi^{\text{IWAE}}\langle \theta, \mathbf{z} \rangle]^{1/2} = \mathcal{O}(K^{-1/2})$
  (Rainforth et al., 2018). This is because $\widehat{\nabla}_\phi^{\text{IWAE}}\langle \theta, \mathbf{z} \rangle$ constitutes a self-normalised
  importance-sampling approximation of $\pi_\theta(\blacktriangledown_\psi - \nabla_\phi \log q_\phi) = 0$, an identity which
  directly follows from Lemma 1 with $f_\psi = w_\psi$.*

- **Inability to achieve zero variance.** *As pointed out in Roeder et al. (2017),
  $\operatorname{var}[\widehat{\nabla}_\phi^{\text{IWAE}}\langle \theta, \mathbf{z} \rangle] > 0$ even in the ideal scenario where $q_\phi = \pi_\theta$ despite the fact that in
  this case, $w_\psi$ is constant and hence $\operatorname{var}[\log \widehat{\mathcal{Z}}_\theta \langle \phi, \mathbf{z} \rangle] = 0$.*

Two modifications of $\widehat{\nabla}_\phi^{\text{IWAE}}\langle \theta, \mathbf{z} \rangle$ have been proposed which (under **R1**) avoid the score-
function terms in (3) and hence (a) exhibit a stable signal-to-noise ratio as $K \to \infty$ and
(b) can achieve zero variance if $q_\phi = \pi_\theta$ (because then $\blacktriangledown_\psi \equiv 0$ since $w_\psi$ is constant).

- **IWAE-STL.** The *'sticking-the-landing' IWAE (IWAE-STL)* gradient proposed by
  Roeder et al. (2017) heuristically ignores the score function terms,

$$\widehat{\nabla}_\phi^{\text{IWAE-STL}}\langle \theta, \mathbf{z} \rangle := \sum_{k=1}^K \bar{w}_\psi^k \, \blacktriangledown_\psi(z^k). \qquad (4)$$

  As shown in Tucker et al. (2019)), this introduces an additional bias whenever $K > 1$.

- **IWAE-DREG.** The *'doubly-reparametrised' IWAE (IWAE-DREG)* gradient proposed by Tucker et al. (2019) removes the score-function terms through Lemma 1,

$$\widehat{\nabla}_\phi^{\text{IWAE-DREG}}\langle\theta, \mathbf{z}\rangle := \sum_{k=1}^{K} \left(\bar{w}_\psi^k\right)^2 \blacktriangledown_\psi(z^k).$$

(5)

The quantities $\widehat{\nabla}_\phi^{\text{IWAE-DREG}}\langle\theta, \mathbf{z}\rangle$ and $\widehat{\nabla}_\phi^{\text{IWAE}}\langle\phi, \mathbf{z}\rangle$ are equal in expectation.

## 2.3 Reweighted wake-sleep (RWS)

The *reweighted wake-sleep (RWS)* algorithm was proposed in Bornschein & Bengio (2015).[1] Letting $\text{KL}(p\|q) := \int_{\mathsf{Z}} \log[p(z)/q(z)]p(z)\,\mathrm{d}z$ is the Kullback–Leibler (KL)-divergence from $p$ to $q$, the RWS algorithm seeks to optimise $\psi = (\theta, \phi)$ as

$$\begin{cases} \theta^\star := \arg\max_\theta \log \mathcal{Z}_\theta, \\ \phi^\star := \arg\min_\phi \text{KL}(\pi_{\theta^\star}\|q_\phi). \end{cases}$$

The $\theta$- and $\phi$-gradients read

$$\begin{bmatrix} \nabla_\theta \log \mathcal{Z}_\theta \\ -\nabla_\phi \text{KL}(\pi_\theta\|q_\phi) \end{bmatrix} = \pi_\theta \begin{pmatrix} \nabla_\theta \log \gamma_\theta \\ \nabla_\phi \log q_\phi \end{pmatrix}.$$

(6)

These quantities are usually intractable and therefore approximated by replacing $\pi_\theta$ by the self-normalised importance sampling approximation $\hat{\pi}_\theta\langle\phi, \mathbf{z}\rangle$ (this does not require **R1**):

$$\begin{bmatrix} \widehat{\nabla}_\theta^{\text{RWS}}\langle\phi, \mathbf{z}\rangle \\ \widehat{\nabla}_\phi^{\text{RWS}}\langle\theta, \mathbf{z}\rangle \end{bmatrix} := \sum_{k=1}^{K} \bar{w}_\psi^k \begin{bmatrix} \nabla_\theta \log \gamma_\theta(z^k) \\ \nabla_\phi \log q_\phi(z^k) \end{bmatrix}.$$

(7)

Since (7) relies on self-normalised importance sampling, Remark 2 shows that its bias relative to (6) is of order $\mathcal{O}(1/K)$. Appendix A discusses the impact of this bias on the $\phi$-gradient in more detail. The optimisation of both $\theta$ and $\phi$ is carried out simultaneously, allowing both gradients to share the same particles and weights. Nonetheless, the lack of a joint objective (for both $\theta$ and $\phi$) is often viewed as the main drawback of RWS.

**RWS-DREG.** Under **R1**, Tucker et al. (2019) proposed the following *'doubly-reparametrised' RWS (RWS-DREG)* gradient which is equal to $\widehat{\nabla}_\phi^{\text{RWS}}\langle\theta, \mathbf{z}\rangle$ in expectation and is derived by applying Lemma 1 to the latter. It reads

$$\widehat{\nabla}_\phi^{\text{RWS-DREG}}\langle\theta, \mathbf{z}\rangle := \sum_{k=1}^{K} \mathcal{F}\left(\bar{w}_\psi^k\right) \blacktriangledown_\psi(z^k),$$

(8)

where the function $\mathcal{F}(w) := w(1-w)$ is used to transform the self-normalised importance weights $\bar{w}_\psi^k$. In high-dimensional settings, it is typically the case that the ordered self-normalised importance weights $\bar{w}_\psi^{(K)} < \ldots < \bar{w}_\psi^{(1)} < 1$ are such that $\bar{w}_\psi^{(1)} \approx 1 - \bar{w}_\psi^{(2)}$ and $\bar{w}_\psi^{(k)} \ll \bar{w}_\psi^{(2)}$ for $k \geq 3$. The transformed weights $\{\mathcal{F}(\bar{w}_\psi^k)\}_{k=1}^K$ are then mainly supported on the *two* particles with the largest self-normalised weights.

## 3 AISLE: A unified adaptive importance-sampling framework

### 3.1 Objective

If $\theta$ is fixed, the RWS algorithm reduces to an adaptive importance-sampling scheme which optimises the proposal distribution by minimising the 'inclusive' KL-divergence from the target distribution $\pi_\theta$ to the proposal $q_\phi$ (see, e.g., Douc et al., 2007; Cappé et al., 2008). If instead $\phi$ is fixed, the RWS algorithm reduces to a stochastic-approximation algorithm for estimating the MLE of the generative-model parameters $\theta$. The advantage of optimising $\theta$

---

[1]Following Tucker et al. (2019) (based on empirical results in Le et al. 2019), we only use the 'wake-phase' $\phi$-updates for RWS.

and $\phi$ simultaneously is that (a) Monte Carlo samples used to approximate the $\theta$-gradient can be re-used to approximate the $\phi$-gradient and (b) optimising $\phi$ typically reduces the error (both in terms of bias and variance) of the $\theta$-gradient approximation.

However, adapting the proposal distribution $q_\phi$ in importance-sampling schemes need not necessarily be based on minimising the (inclusive) KL-divergence. Numerous other techniques exist in the literature (e.g. Geweke, 1989; Evans, 1991; Oh & Berger, 1992; Richard & Zhang, 2007; Cornebise et al., 2008) and may sometimes be preferable. Indeed, another popular approach with strong theoretical support is based on minimising the $\chi^2$-divergence (see, e.g., Deniz Akyildiz & Míguez, 2019). Based on this insight, we slightly generalise the RWS-objective as

$$\begin{cases} \theta^\star \coloneqq \arg\max_\theta \log \mathcal{Z}_\theta, \\ \phi^\star \coloneqq \arg\min_\phi \mathrm{D_f}(\pi_{\theta^\star}\|q_\phi). \end{cases} \tag{9}$$

Here, $\mathrm{D_f}(p\|q) \coloneqq \int_\mathsf{Z} \mathrm{f}(p(z)/q(z))q(z)\,\mathrm{d}z$ is some f-divergence from $p$ to $q$. We reiterate that alternative approaches for optimising $\phi$ (which do not minimise f-divergences) could be used. However, we state (9) for concreteness as it suffices for the remainder of this work; we call the resulting algorithm *adaptive importance sampling for learning (AISLE)*. As will become clear below, this unified framework permits a straightforward and principled derivation of robust $\phi$-gradient estimators that do not degenerate as $K \to \infty$.

## 3.2 $\theta$-GRADIENT

Optimisation is again performed via a stochastic gradient-ascent. The intractable $\theta$-gradient $\nabla_\theta \log \mathcal{Z}_\theta = \pi_\theta(\nabla_\theta \log \gamma_\theta)$ is approximated as in RWS, i.e. for $\mathbf{z} \sim q_\phi^{\otimes K}$:

$$\widehat{\nabla}_\theta^{\mathrm{AISLE}}\langle\phi, \mathbf{z}\rangle \coloneqq \widehat{\nabla}_\theta^{\mathrm{RWS}}\langle\phi, \mathbf{z}\rangle = \widehat{\nabla}_\theta^{\mathrm{IWAE}}\langle\phi, \mathbf{z}\rangle = \sum_{k=1}^K \bar{w}_\psi^k \nabla_\theta \log \gamma_\theta(z^k).$$

The $\theta$-gradient is thus the same for all algorithms discussed in this work although the IWAE-paradigm views it as an unbiased gradient of a (biased) lower-bound to the evidence, while AISLE (and RWS) interpret it as a self-normalised importance-sampling (and consequently biased) approximation of the gradient $\nabla_\theta \log \mathcal{Z}_\theta$ for the 'exact' objective.

## 3.3 $\phi$-GRADIENT

### 3.3.1 GENERAL DERIVATION

In the derivations to follow, integrals of the form $\pi_\theta([F \circ w_\psi]\nabla_\phi \log q_\phi)$ naturally appear. These can also be expressed as $\mathcal{Z}_\theta^{-1} q_\phi([H \circ w_\psi]\nabla_\phi \log q_\phi)$ with $H(y) \coloneqq F(y)y$. By Lemma 1,

$$\pi_\theta([F \circ w_\psi]\nabla_\phi \log q_\phi) = \frac{1}{\mathcal{Z}_\theta} \mathbb{E}_{z \sim q_\phi}\left[w_\psi H'(w_\psi(z))\blacktriangledown_\psi(z)\right].$$

Approximating the expectation as well as the normalising constant $\mathcal{Z}_\theta$ on the r.h.s. with the vanilla Monte Carlo method with $K$ samples $z^1, \ldots, z^K \overset{\mathrm{iid}}{\sim} q_\phi^{\otimes K}$ yields the estimator

$$\pi_\theta([F \circ w_\psi]\nabla_\phi \log q_\phi) \approx \sum_{k=1}^K \bar{w}_\psi^k H'(w_\psi(z^k))\,\blacktriangledown_\psi(z^k). \tag{10}$$

Remark 2 shows that this approximation has a bias of order $\mathcal{O}(K^{-1})$ and a standard-deviation of order $\mathcal{O}(K^{-1/2})$. Now, most of the f-divergences used for variational inference in intractable models are such that there exists a function $\tilde{f}\colon \mathbb{R} \to \mathbb{R}$ satisfying $\mathrm{D_f}(\pi_\theta\|q_\phi) = \mathcal{Z}_\theta^\kappa \int_\mathsf{Z} \tilde{f}[w_\psi(z)]q_\phi(z)\,\mathrm{d}z + C(\theta)$ for an exponent $\kappa \in \mathbb{R}$ and constant $C(\theta)$ independent of $\phi$. In other words, for a given value of $\theta$, the optimization of the f-divergence as a function of $\phi$ can be carried out without relying on the knowledge of $\mathcal{Z}_\theta$. Writing $g(y) \coloneqq \tilde{f}'(y) - \tilde{f}(y)/y$, simple algebra then directly shows that

$$-\nabla_\phi \mathrm{D_f}(\pi_\theta\|q_\phi) = \mathcal{Z}_\theta^{\kappa+1} \int_\mathsf{Z} g(w_\psi(z))[\nabla_\phi \log q_\phi(z)]\pi_\theta(z)\,\mathrm{d}z. \tag{11}$$

Since the integral in (11) is an expectation with respect to $\pi_\theta$, it can be approximated with self-importance sampling, possibly multiplied an additional importance-sampling approximation $\widehat{\mathcal{Z}}_\theta\langle\phi, \mathbf{z}\rangle$ of $\mathcal{Z}_\theta$ raised to some power. This leads to,

$$-\nabla_\phi \, \mathrm{D_f}(\pi_\theta \| q_\phi) \approx \widehat{\mathcal{Z}}_\theta\langle\phi, \mathbf{z}\rangle^{\kappa+1} \sum_{k=1}^{K} \bar{w}_\psi^k g(w_\psi(z^k)) \nabla_\phi \log q_\phi(z^k). \tag{12}$$

Indeed, Equation (10) applies to (11), leading to the reparametrised estimator

$$-\nabla_\phi \, \mathrm{D_f}(\pi_\theta \| q_\phi) \approx \widehat{\mathcal{Z}}_\theta\langle\phi, \mathbf{z}\rangle^{\kappa+1} \sum_{k=1}^{K} \bar{w}_\psi^k \, h'(w_\psi(z^k)) \blacktriangledown_\psi(z^k), \tag{13}$$

where $h(y) = g(y)y$ and $g \colon \mathbb{R} \to \mathbb{R}$ given immediately above (11). We now describe several particular cases.

### 3.3.2 SPECIAL CASE 'INCLUSIVE' KL-DIVERGENCE: RWS AND IWAE-STL

We have $\mathrm{KL}(\pi_\theta \| q_\phi) = \mathcal{Z}_\theta^\kappa \int_{\mathsf{Z}} \tilde{f}(w_\psi(z)) q_\phi(z) \, \mathrm{d}z + C(\theta)$ with $\kappa = -1$ and $\tilde{f}(y) = y \log(y)$. In that case, with the notations of Section 3.3.1, we have $g(y) = 1$ and $h'(y) = 1$.

- **AISLE-KL-NOREP/RWS.** Without relying on any reparametrisation, Equation (12) yields the following gradient, which clearly equals $\widehat{\nabla}_\phi^{\text{RWS}}\langle\theta, \mathbf{z}\rangle$:

$$-\nabla_\phi \, \mathrm{D_f}(\pi_\theta \| q_\phi) \approx \widehat{\nabla}_\phi^{\text{AISLE-KL-NOREP}}\langle\theta, \mathbf{z}\rangle := \sum_{k=1}^{K} \bar{w}_\psi^k \, \nabla_\phi \log q_\phi(z^k). \tag{14}$$

- **AISLE-KL.** Using reparametrisation, Equation (13) yields the gradient:

$$-\nabla_\phi \, \mathrm{D_f}(\pi_\theta \| q_\phi) \approx \widehat{\nabla}_\phi^{\text{AISLE-KL}}\langle\theta, \mathbf{z}\rangle := \sum_{k=1}^{K} \bar{w}_\psi^k \, \blacktriangledown_\psi(z^k). \tag{15}$$

We thus arrive at the following result which demonstrates that IWAE-STL can be derived in a principled manner from AISLE, i.e. without the need for a multi-sample objective.

**Proposition 1.** *For any* $(\theta, \phi, \mathbf{z})$, $\widehat{\nabla}_\phi^{\text{AISLE-KL}}\langle\theta, \mathbf{z}\rangle = \widehat{\nabla}_\phi^{\text{IWAE-STL}}\langle\theta, \mathbf{z}\rangle$.

Proposition 1 is notable because it shows that IWAE-STL (which avoids the breakdown highlighted in Rainforth et al. (2018) and which can also achieve zero variance) can be derived in a principled manner from AISLE, i.e. without relying on a multi-sample objective. Proposition 1 thus provides a theoretical basis for IWAE-STL which was previously viewed as an alternative gradient for IWAE for which it is biased and only heuristically justified. Furthermore, the fact that IWAE-STL exhibited good empirical performance in Tucker et al. (2019) even in an example in which RWS broke down, suggests that this breakdown may not be due to RWS' lack of optimising a joint objective as previously conjectured.

Finally, recall that Tucker et al. (2019) obtained an alternative 'doubly-reparametrised' RWS $\phi$-gradient $\widehat{\nabla}_\phi^{\text{RWS-DREG}}\langle\theta, \mathbf{z}\rangle$ given in (8) by *first* replacing the exact (but intractable) $\phi$-gradient by the self-normalised importance-sampling approximation $\widehat{\nabla}_\phi^{\text{RWS}}\langle\theta, \mathbf{z}\rangle$ and *then* applying the identity from Lemma 1. Note that this may result in a variance reduction but does not change the bias of the gradient estimator. In contrast, AISLE-KL is derived by *first* applying Lemma 1 to the exact (RWS) $\phi$-gradient and *then* approximating the resulting expression. This can potentially reduce both bias and variance.

### 3.3.3 SPECIAL CASE $\alpha$-DIVERGENCE: IWAE-DREG

Up to some irrelevant additive constant, the $\alpha$-divergence between two distributions $p$ and $q$ is given by $\int_{\mathsf{Z}} (p(z)/q(z))^\alpha q(z) \, \mathrm{d}z$ for some $\alpha > 1$. This can also be expressed as $\mathcal{Z}_\theta^\kappa \int_{\mathsf{Z}} \tilde{f}(w_\psi(z)) q_\phi(z) \, \mathrm{d}z$ with $\kappa = -\alpha$ and $\tilde{f}(y) = y^\alpha$. In this case, with the notation from Section 3.3.1, we have $g(y) = (\alpha - 1)y^{\alpha-1}$ and $h'(y) = \alpha(\alpha - 1) \, y^{\alpha-1}$. Note that the case $\alpha = 2$ is equivalent, up to an irrelevant additive constant, to a standard $\chi^2$-divergence. Minimising this divergence is natural in importance sampling since $\chi^2(\pi_\theta \| q_\phi) = \mathrm{var}_{z \sim q_\phi}[w_\psi/\mathcal{Z}_\theta]$ is the variance of the importance weights.

- **AISLE-$\alpha$-NOREP.** Without relying on any reparametrisation, Equation (13) yields

$$\widehat{\nabla}_\phi^{\text{AISLE-}\alpha\text{-NOREP}}\langle\theta, \mathbf{z}\rangle := (\alpha - 1)K^{\alpha-1}\sum_{k=1}^{K}(\bar{w}_\psi^k)^\alpha \nabla_\phi \log q_\phi(z^k), \quad (16)$$

  with the following special case which is also proportional to the 'score gradient' from Dieng et al. (2017, Appendix G): $\widehat{\nabla}_\phi^{\text{AISLE-}\chi^2\text{-NOREP}}\langle\theta, \mathbf{z}\rangle := K\sum_{k=1}^{K}(\bar{w}_\psi^k)^2 \nabla_\phi \log q_\phi(z^k)$.

- **AISLE-$\alpha$.** Using reparametrisation, Equation (12) becomes

$$\widehat{\nabla}_\phi^{\text{AISLE-}\alpha}\langle\theta, \mathbf{z}\rangle := \alpha(\alpha - 1)K^{\alpha-1}\sum_{k=1}^{K}(\bar{w}_\psi^k)^\alpha \blacktriangledown_\psi(z^k), \quad (17)$$

  again with the special case $\widehat{\nabla}_\phi^{\text{AISLE-}\chi^2}\langle\theta, \mathbf{z}\rangle := 2K\sum_{k=1}^{K}(\bar{w}_\psi^k)^2 \blacktriangledown_\psi(z^k)$.

This demonstrates that IWAE-DREG can be derived (up to the proportionality factor $2K$) in a principled manner from AISLE, i.e. without the need for a multi-sample objective.

**Proposition 2.** *For any* $(\theta, \phi, \mathbf{z})$, $\widehat{\nabla}_\phi^{\text{AISLE-}\chi^2}\langle\theta, \mathbf{z}\rangle = 2K\widehat{\nabla}_\phi^{\text{IWAE-DREG}}\langle\theta, \mathbf{z}\rangle$. $\qquad\square$

Note that if the implementation normalises the gradients, e.g. as effectively done by ADAM (Kingma & Ba, 2015), the constant factor cancels out and AISLE-$\chi^2$ becomes equivalent to IWAE-DREG. Otherwise (e.g. in plain stochastic gradient-ascent) this shows that the learning rate needs to be scaled as $\mathcal{O}(K)$ for the IWAE or IWAE-DREG $\phi$-gradients.

### 3.3.4 SPECIAL CASE 'EXCLUSIVE' KL-DIVERGENCE.

For the 'exclusive' KL-divergence, we have $\text{KL}(q_\phi\|\pi_\theta) = \int \tilde{f}(w_\psi(z))q_\phi(z)\,\mathrm{d}z + C(\theta)$ with $\tilde{f}(y) = \log(y)$. In that case, with the notation from Section 3.3.1, we have $h'(y) = 1/y$. This directly leads to the following approximation,

$$-\nabla_\phi \text{D}_\text{f}(\pi_\theta\|q_\phi) \approx \widehat{\nabla}_\phi^{\text{AISLE-REV-KL}}\langle\theta, \mathbf{z}\rangle := \frac{1}{K}\sum_{k=1}^{K}\blacktriangledown_\psi(z^k).$$

This can be recognised as a simple average over $K$ independent replicates of the 'sticking-the-landing' estimator for VAEs proposed in Roeder et al. (2017, Equation 8). As we discuss in Appendix A, optimising this 'exclusive' KL-divergence can sometimes lead to faster convergence of $\phi$ than optimising the 'inclusive' KL-divergence $\text{KL}(\pi_\theta\|q_\phi)$. However, care must be taken because minimising the exclusive divergence does not necessarily lead to well behaved or even well-defined importance weights and thus can negatively affect learning of $\theta$ (whose gradient is a self-normalised importance-sampling approximation which makes use of those weights).

## 4 CONCLUSION

We have shown that the adaptive-importance sampling paradigm of the *reweighted wake-sleep (RWS)* (Bornschein & Bengio, 2015) is preferable to the multi-sample objective paradigm of *importance weighted autoencoders (IWAEs)* (Burda et al., 2016) because the former achieves all the goals of the latter whilst avoiding its drawbacks. To formalise this argument, we have introduced a simple, unified adaptive-importance-sampling framework termed *adaptive importance sampling for learning (AISLE)* (which slightly generalises the RWS algorithm) and have proved that AISLE allows us to derive the *'sticking-the-landing' IWAE (IWAE-STL)* gradient from Roeder et al. (2017) and the *'doubly-reparametrised' IWAE (IWAE-DREG)* gradient from Tucker et al. (2019) as special cases.

We hope that this work highlights the potential for further improving variational techniques by drawing upon the vast body of research on (adaptive) importance sampling in the computational statistics literature. Conversely, the methodological connections established in this work may also serve to emphasise the utility of the reparametrisation trick from Kingma & Welling (2014); Tucker et al. (2019) to computational statisticians.

In a companion article, we are extending the present work to the *variational sequential Monte Carlo* methods from Maddison et al. (2017); Le et al. (2018); Naesseth et al. (2018) and to the *tensor Monte Carlo* approach from Aitchison (2018).

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

## A  ON THE RÔLE OF THE SELF-NORMALISATION BIAS WITHIN RWS/AISLE

### A.1  THE SELF-NORMALISATION BIAS

Within the self-normalised importance-sampling approximation, the number of particles, $K$, interpolates between two extremes:

- As $K \uparrow \infty$, $\hat{\pi}_\theta \langle \phi, \mathbf{z} \rangle(f)$ becomes an increasingly accurate approximation of $\pi_\theta(f)$.
- For $K = 1$, however, $\hat{\pi}_\theta \langle \phi, \mathbf{z} \rangle(f) = f(z^1)$ reduces to a vanilla Monte Carlo approximation of $q_\phi(f)$ (because the single self-normalised importance weight is always equal to 1).

This leads to the following insight about the estimators $\widehat{\nabla}_\phi^{\text{AISLE-KL}} \langle \theta, \mathbf{z} \rangle$ and $\widehat{\nabla}_\phi^{\text{AISLE-}\chi^2} \langle \theta, \mathbf{z} \rangle$.

- As $K \uparrow \infty$, these two estimators become increasingly accurate approximations of the *'inclusive'*-divergence gradients $-\nabla_\phi \text{KL}(\pi_\theta \| q_\phi) = \pi_\theta(\blacktriangledown_\phi)$ and $-\nabla_\phi \chi^2(\pi_\theta \| q_\phi) = 2\pi_\theta([w_\psi/\mathcal{Z}_\theta]\blacktriangledown_\phi)$, respectively.
- For $K = 1$, however, these two estimators reduce to vanilla Monte Carlo approximations of the *'exclusive'*-divergence gradients $-\nabla_\phi \text{KL}(q_\phi \| \pi_\theta) = q_\phi(\blacktriangledown_\phi)$ and $-2\nabla_\phi \text{KL}(q_\phi \| \pi_\theta) = 2q_\phi(\blacktriangledown_\phi)$, respectively.

This is similar to the standard IWAE $\phi$-gradient which also represents a vanilla Monte Carlo approximation of $-\nabla_\phi \text{KL}(q_\phi \| \pi_\theta)$ if $K = 1$ as IWAE reduces to a VAE in this case.

Characterising the small-$K$ self-normalisation bias of the reparametrisation-free AISLE $\phi$ gradients, AISLE-KL-NOREP and AISLE-$\chi^2$-NOREP, is more difficult because if $K = 1$, they constitute vanilla Monte Carlo approximations of $q_\phi(\nabla_\phi \log q_\phi) = 0$. Nonetheless, Le et al. (2019, Figure 5) lends some support to the hypothesis that the small-$K$ self-normalisation bias of these gradients also favours a minimisation of the exclusive KL-divergence.

### A.2  INCLUSIVE VS EXCLUSIVE KL-DIVERGENCE MINIMISATION

Recall that the main motivation for use of IWAEs (instead of VAEs) was the idea that we could use self-normalised importance-sampling approximations with $K > 1$ particles to reduce the bias of the $\theta$-gradient relative to $\nabla_\theta \log \mathcal{Z}_\theta$. The error of such (self-normalised) importance-sampling approximations can be controlled by ensuring that $q_\phi$ is close to $\pi_\theta$ (in some suitable sense) in any part of the space $\mathsf{Z}$ in which $\pi_\theta$ has positive probability mass. For instance, it is well known that the error will be small if the 'inclusive' KL-divergence $\text{KL}(\pi_\theta \| q_\phi)$ is small as this implies well-behaved importance weights. In contrast, a small 'exclusive' KL-divergence $\text{KL}(q_\phi \| \pi_\theta)$ is not sufficient for well-behaved importance weights because the latter only ensures that $q_\phi$ is close to $\pi_\theta$ in those parts of the space $\mathsf{Z}$ in which $q_\phi$ has positive probability mass.

Let $\mathcal{Q} := \{q_\phi\}$ (which is indexed by $\phi$) be the family of proposal distributions/the variational family. Then we can distinguish two scenarios.

1. **Sufficiently expressive $\mathcal{Q}$.** For the moment, assume that the family $\mathcal{Q}$ is flexible ('expressive') enough in the sense that it contains a distribution $q_{\phi^\star}$ which is (at least approximately) equal to $\pi_\theta$ and that our optimiser can reach the value $\phi^\star$ of $\phi$. In this case, minimising the exclusive KL-divergence can still yield well-behaved importance weights because in this case, $\phi^\star := \arg\min_\phi \text{KL}(\pi_\theta \| q_\phi)$ is (at least approximately) equal to $\arg\min_\phi \text{KL}(q_\phi \| \pi_\theta)$.

2. **Insufficiently expressive $\mathcal{Q}$.** In general, the family $\mathcal{Q}$ is not flexible enough in the sense that all of its members are 'far away' from $\pi_\theta$, e.g. if the $D$ components $z_1, \ldots, z_D$ of $z = z_{1:D}$ are highly correlated under $\pi_\theta$ whilst $q_\phi(z) = \prod_{d=1}^D q_{\phi,d}(z_d)$ is fully factorised. In this case, minimising the exclusive KL-divergence could lead to poorly-behaved importance weights and we should optimise $\phi^\star := \arg\min_\phi \text{KL}(\pi_\theta \| q_\phi)$ as discussed above.

**Remark 4.** *In Scenario 1 above, i.e. for a sufficiently flexible $\mathcal{Q}$, using a gradient-descent algorithm which seeks to minimise the* exclusive *divergence can sometimes be preferable to a gradient-descent algorithm which seeks to minimise the* inclusive *divergence. This is because both find (approximately) the same optimum but the latter may exhibit faster convergence in some applications. In such scenarios, the discussion in Subsection A.1 indicates that a smaller number of particles, $K$, could then be preferable for some of the $\phi$-gradients because (a) the $\mathcal{O}(K^{-1})$ self-normalisation bias outweighs the $\mathcal{O}(K^{-1/2})$ standard deviation and (b) the direction of this bias may favour faster convergence.*

Unfortunately, simply setting $K = 1$ for the approximation of the $\phi$-gradients[2] is not necessarily optimal because

- even in the somewhat idealised scenario 1 above and even if the direction of the self-normalisation bias encourages faster convergence, increasing $K$ is still desirable to reduce the variance of the gradient approximations and furthermore, even in this scenario, seeking to optimise the exclusive KL-divergence could lead to poorly behaved importance-sampling approximations of the $\theta$-gradient whenever $\phi$ is still far away from optimal;

- not using the information contained in *all $K$* particles and weights (which have already been sampled/calculated to approximate the $\theta$-gradient) seems wasteful;

- if $K = 1$, the reparametrisation-free AISLE $\phi$-gradients, AISLE-KL-NOREP and AISLE-$\chi^2$-NOREP are simply vanilla Monte Carlo estimates of 0 and the RWS-DREG $\phi$-gradient is then equal to 0.

## B EMPIRICAL ILLUSTRATION

### B.1 ALGORITHMS

In these supplementary materials, we illustrate the different $\phi$-gradient estimators (recall that all algorithms discussed in this work share the same $\theta$-gradient estimator). Specifically, we compare the following approximations.

- **AISLE-KL-NOREP.** The gradient for AISLE based on the KL-divergence without any further reparametrisation from (14) i.e. this coincides with the standard RWS-gradient from (7). This gradient does not require **R1** but does not achieve zero variance even if $q_\phi = \pi_\theta$.

- **AISLE-KL.** The gradient for AISLE based on the KL-divergence after reparametrising and exploiting the identity from Lemma 1; it is given by (15) and coincides with the IWAE-STL-gradient from (4).

- **AISLE-$\chi^2$-NOREP.** The gradient for AISLE based on the $\chi^2$-divergence without any reparametrisation given in (16). This gradient again does not require **R1** but does not achieve zero variance even if $q_\phi = \pi_\theta$.

- **AISLE-$\chi^2$.** The gradient for AISLE based on the $\chi^2$-divergence after reparametrising and exploiting the identity from Lemma 1; it is given by (17) and is alsow proportional to IWAE-DREG from Tucker et al. (2019) which was stated in (5). When normalising the gradients (as, e.g. implicitly done by optimisers such as ADAM Kingma & Ba, 2015) the proportionality constant cancels out so that both these gradient approximations lead to computationally the same algorithm.

- **IWAE.** The gradient for IWAE employing the reparametrisation trick from Kingma & Welling (2014). Its sampling approximation is given in (3). Recall that this is the $\phi$-gradient whose signal-to-noise ratio degenerates with $K$ as pointed out in Rainforth et al. (2018) (and which also cannot achieve zero variance even if $q_\phi = \pi_\theta$).

---

[2]Within the IWAE-paradigm, using different numbers of particles for the $\theta$ and $\phi$-gradients was recently proposed in Rainforth et al. (2018); Le et al. (2018) who termed this approach *'alternating evidence lower bounds'*, albeit their aim was to circumvent the signal-to-noise ratio breakdown of the IWAE $\phi$-gradient which is distinct from the phenomenon discussed here.

- **IWAE-DREG.** The 'doubly-reparametrised' IWAE gradient from (5) which was proposed in Tucker et al. (2019). It is proportional to AISLE-$\chi^2$.

- **RWS-DREG.** The 'doubly-reparametrised' RWS $\phi$-gradient from (8) which was proposed in Tucker et al. (2019) who derived it by applying the identity from Lemma 1 to the RWS $\phi$-gradient.

## B.2 Model

**Generative model.** We have $N$ $D$-dimensional observations $x^{(1)}, \ldots, x^{(N)} \in \mathbb{R}^D$ and $N$ $D$-dimensional latent variables $z^{(1)}, \ldots, z^{(N)} \in \mathbb{R}^D$. Unless otherwise stated, any vector $y \in \mathbb{R}^D$ is to be viewed as a $D \times 1$ column vector.

Hereafter, wherever necessary, we add an additional subscript to make the dependence on the observations explicit. The joint law (the 'generative model'), parametrised by $\theta$, of the observations and latent variables then factorises as

$$\prod_{n=1}^{N} p_\theta(z^{(n)}) p_\theta(x^{(n)}|z^{(n)}) = \prod_{n=1}^{N} \gamma_{\theta, x^{(n)}}(z^{(n)}).$$

We model each latent variable–observation pair $(z, x)$ as

$$p_\theta(z) \coloneqq \mathrm{N}(z; \mu, \Sigma),$$
$$p_\theta(x|z) \coloneqq \mathrm{N}(x; z; \mathrm{I}),$$

where $\theta \coloneqq \mu = \mu_{1:D} \in \mathbb{R}^D$, where $\Sigma \coloneqq (\sigma_{d,d'})_{(d,d') \in \{1, \ldots, D\}} \in \mathbb{R}^{D \times D}$ is assumed to be known and where I denotes the $D \times D$-identity matrix. For any $\theta$,

$$\mathcal{Z}_{\theta, x} = p_\theta(x) = \mathrm{N}(x; \mu, \mathrm{I} + \Sigma), \tag{18}$$
$$\pi_{\theta, x}(z) = p_\theta(z|x) = \mathrm{N}(z; \nu_{\theta, x}, P), \tag{19}$$

with $P \coloneqq (\Sigma^{-1} + \mathrm{I})^{-1}$ and $\nu_{\theta, x} \coloneqq P(\Sigma^{-1}\mu + x)$. In particular, (18) implies that $\theta^{\mathrm{ML}} = \frac{1}{N} \sum_{n=1}^{N} x^{(n)}$.

**Proposal/variational approximation.** We take the proposal distributions as a fully-factored Gaussian:

$$q_{\phi, x}(z) \coloneqq \mathrm{N}(z; Ax + b, C), \tag{20}$$

where $A = (a_{d,d'})_{(d,d') \in \{1, \ldots, D\}^2} \in \mathbb{R}^{D \times D}$, $b = b_{1:D} \in \mathbb{R}^D$ and, for $c_{1:D} =: c \in \mathbb{R}^D$, $C \coloneqq \mathrm{diag}(\mathrm{e}^{2c_1}, \ldots, \mathrm{e}^{2c_D})$. The parameters to optimise are thus

$$\phi \coloneqq (a_1^{\mathrm{T}}, \ldots, a_D^{\mathrm{T}}, b^{\mathrm{T}}, c^{\mathrm{T}}),$$

where $a_d \coloneqq [a_{d,1}, a_{d,2}, \ldots, a_{d,D}]^{\mathrm{T}} \in \mathbb{R}^{D \times 1}$ denotes the column vector formed by the elements in the $d$th row of $A$. Furthermore, for the reparametrisation trick, we take $q(\epsilon) \coloneqq \mathrm{N}(\epsilon; 0, \mathrm{I})$, where $0 \in \mathbb{R}^D$ is a vector whose elements are all 0, so that

$$h_{\phi, x}(\epsilon) \coloneqq Ax + b + C^{1/2}\epsilon,$$

which means that $h_{\phi, x}^{-1}(z) = C^{-1/2}(z - Ax - b)$.

Note that the mean of the proposal in (20) coincides with the mean of the posterior in (19) if $A = P$ and $b = P\Sigma^{-1}\mu$.

This model is similar to the one used as a benchmark in Rainforth et al. (2018, Section 4) and also in Tucker et al. (2019, Section 6.1) who specified both the generative model and the variational approximation to be isotropic Gaussians. Specifically, their setting can be recovered by taking $\Sigma \coloneqq \mathrm{I}$ and fixing $c_d = \log(2/3)/2$ so that $C = \frac{2}{3}\mathrm{I}$ throughout. Here, in order to investigate a slightly more realistic scenario, we also allow for the components of the latent vectors $z$ to be *correlated/dependent* under the generative model. However, as the variational approximation remains restricted to being fully factored, it may fail to fully capture the uncertainty about the latent variables.

**Gradient calculations.** We end this subsection by stating the expressions needed to calculate the gradients in the Gaussian example presented above. Throughout, we use the *denominator-layout* notation for vector and matrix calculus and sometimes write $\epsilon = \epsilon_{1:D} = h_{\phi,x}^{-1}(z)$ to simplify the notation. Thus,

$$\nabla_\theta \log \gamma_{\theta,x}(z) = \Sigma^{-1}(z - \mu) \in \mathbb{R}^D,$$
$$\nabla_z \log \gamma_{\theta,x}(z) = \Sigma^{-1}(\mu - z) + x - z \in \mathbb{R}^D, \tag{21}$$
$$\nabla_z \log q_{\phi,x}(z) = -C^{-1}(z - Ax - b)$$
$$= -C^{-1/2}\epsilon \in \mathbb{R}^D. \tag{22}$$

Let $a_d := [a_{d,1}, a_{d,2}, \ldots, a_{d,D}]^{\mathrm{T}} \in \mathbb{R}^{D \times 1}$ denote the column vector formed by the elements in the $d$th row of $A$. Then, letting $\odot$ denote elementwise multiplication and using the convention that addition or subtraction of the scalar 1 is to be done elementwise,

$$\nabla_{a_d} \log q_{\phi,x}(z) = \exp(-2c_d)(z_d - a_d^{\mathrm{T}}x - b_d)x$$
$$= \exp(-c_d)\epsilon_d x \in \mathbb{R}^D, \quad d \in \{1, \ldots, D\},$$
$$\nabla_b \log q_{\phi,x}(z) = C^{-1}(z - Ax - b)$$
$$= C^{-1/2}\epsilon \in \mathbb{R}^D,$$
$$\nabla_c \log q_{\phi,x}(z) = C^{-1/2}(z - Ax - b) \odot C^{-1/2}(z - Ax - b) - 1$$
$$= \epsilon \odot \epsilon - 1 \in \mathbb{R}^D,$$

Furthermore, write $h_{\phi,x} = [h_{\phi,x,1}, \ldots, h_{\phi,x,D}]^{\mathrm{T}}$, i.e.

$$h_{\phi,x,d}(\epsilon) = z_d = a_d^{\mathrm{T}}x + b_d + \exp(c_d)\epsilon_d,$$

and let $\iota^{(d)} = [0, \ldots, 0, 1, 0, \ldots, 0]^{\mathrm{T}} \in \mathbb{R}^D$ be the vector whose entries are all 0 except for the $d$th entry which is 1. Then, for $d \in \{1, \ldots, D\}$,

$$[\nabla_{a_{d'}} h_{\phi,x,d}](\epsilon) = \mathbf{1}\{d = d'\}x \in \mathbb{R}^D, \quad d' \in \{1, \ldots, D\}, \tag{23}$$
$$[\nabla_b h_{\phi,x,d}](\epsilon) = \iota^{(d)} \in \mathbb{R}^D, \tag{24}$$
$$[\nabla_c h_{\phi,x,d}](\epsilon) = \exp(c_d)\epsilon_d \iota^{(d)} \in \mathbb{R}^D. \tag{25}$$

Again writing $\epsilon = h_{\phi,x}^{-1}(z)$ implies that

$$\nabla_\phi[\log \circ w_{\psi',x} \circ h_{\phi,x}]|_{\psi'=\psi}(\epsilon) = [\nabla_\phi h_{\phi,x,1}, \ldots, \nabla_\phi h_{\phi,x,D}](\epsilon)\nabla_z \log w_{\psi,x}(z),$$

so that, letting $[\nabla_z \log w_{\psi,x}(z)]_d$ denote the $d$th element of the vector $\nabla_z \log w_{\psi,x}(z)$,

$$\nabla_{a_d}[\log \circ w_{\psi',x} \circ h_{\phi,x}]|_{\psi'=\psi}(\epsilon) = [\nabla_z \log w_{\psi,x}(z)]_d x,$$
$$\nabla_b[\log \circ w_{\psi',x} \circ h_{\phi,x}]|_{\psi'=\psi}(\epsilon) = \nabla_z \log w_{\psi,x}(z),$$
$$\nabla_c[\log \circ w_{\psi',x} \circ h_{\phi,x}]|_{\psi'=\psi}(\epsilon) = \epsilon \odot C^{1/2}\nabla_z \log w_{\psi,x}(z).$$

From this, since

$$\nabla_\phi[\log \circ w_{\psi,x} \circ h_{\phi,x}](\epsilon) = \nabla_\phi[\log \circ w_{\psi',x} \circ h_{\phi,x}]|_{\psi'=\psi}(\epsilon) - \nabla_\phi \log q_{\phi,x}(z),$$

we have that

$$\nabla_{a_d}[\log \circ w_{\psi,x} \circ h_{\phi,x}](\epsilon) = ([\nabla_z \log w_{\psi,x}(z)]_d - C^{-1/2}\epsilon_d)x,$$
$$\nabla_b[\log \circ w_{\psi,x} \circ h_{\phi,x}](\epsilon) = \nabla_z \log w_{\psi,x}(z) - C^{-1/2}\epsilon,$$
$$\nabla_c[\log \circ w_{\psi,x} \circ h_{\phi,x}](\epsilon) = \epsilon \odot C^{1/2}\nabla_z \log w_{\psi,x}(z) - \epsilon \odot \epsilon + 1.$$

**Impact of the reparametrisation.** We end this subsection by briefly illustrating the impact of the reparametrisation trick combined with the identity from Tucker et al. (2019) which was given in Lemma 1. Recall that this approach yields $\phi$-gradients that are expressible as integrals of path-derivative functions $\blacktriangledown_{\psi,x} := \nabla_\phi[\log \circ w_{\psi',x} \circ h_{\phi,x}]|_{\psi'=\psi} \circ h_{\phi,x}^{-1}$, Thus, if there exists a value $\phi$ such that $q_{\phi,x} = \pi_{\theta,x}$ then $w_{\psi,x} \propto \pi_{\theta,x}/q_{\phi,x} \equiv 1$ is constant so that we obtain zero-variance $\phi$-gradients (see, e.g., Roeder et al., 2017, for a discussion on this).

For simplicity, assume that $\Sigma = I$ and recall that we then have $q_{\phi^\star,x} = \pi_{\theta,x}$ if the values $(A, b, C)$ implied by $\phi^\star$ are $(A^\star, b^\star, C^\star) = (\frac{1}{2}I, \frac{1}{2}\mu, \frac{1}{2}I)$.

By (21) and (22), and with the usual convention $\epsilon = h_{\phi,x}^{-1}(z)$, we then have

$$\begin{aligned}
\nabla_z \log w_{\psi,x}(z) &= (x + \mu) - 2z + C^{-1}(z - Ax - b) \\
&= 2[(A^\star x + b^\star) - (Ax + b) + C^{-1/2}(C^\star - C)\epsilon].
\end{aligned} \quad (26)$$

Note that the only source of randomness in this expression is the multivariate normal random variable $\epsilon$. Thus, by (23) and (24), for *any* values of $A$ and $b$ and *any* $K \geq 1$, the variance of the $A$- and $b$-gradient portion of AISLE-KL/IWAE-STL and AISLE-$\chi^2$/IWAE-DREG goes to zero as $C \to C^\star = \frac{1}{2}I$. In other words, in this model, these 'score-function free' $\phi$-gradients achieve (near) zero variance for the parameters governing the proposal mean as soon as the variance-parameters fall within a neighbourhood of their optimal values. Furthermore, (25) combined with (26) shows that for *any* $K \geq 1$, the variance of the $C$-gradient portion also goes to zero as $(A, b, C) \to (A^\star, b^\star, C^\star)$. A more thorough analysis of the benefits of reparametrisation-trick gradients in Gaussian settings is carried out in Xu et al. (2019).

### B.3 SIMULATIONS

**Setup.** We end this section by empirically comparing the algorithms from Subsection B.1. We run each of these algorithms for a varying number of particles, $K \in \{1, 10, 100\}$, and varying model dimensions, $D \in \{2, 5, 10\}$. Each of these configurations is repeated independently 100 times. Each time using a new synthetic data set consisting of $N = 25$ observations sampled from the generative model after generating a new 'true' prior mean vector as $\mu \sim N(0, I)$. Since all the algorithms share the same $\theta$-gradient, we focus only on the optimisation of $\phi$ and thus simply fix $\theta := \theta^{\text{ML}}$ throughout. We show results for the following model settings.

- **Figure 1.** The generative model is specified via $\Sigma = I$. In this case, there exists a value $\phi^\star$ of $\phi$ such that $q_{\phi,x}(z) = \pi_{\theta,x}(z)$. Note that this corresponds to Scenario 1 in Subsection A.2.

- **Figure 2.** The generative model is specified via $\Sigma = (0.95^{|d-d'|+1})_{(d,d') \in \{1,\dots,D\}^2}$. Note that in this case, the fully-factored variational approximation cannot fully mimic the dependence structure of the latent variables under the generative model. That is, in this case, $q_{\phi,x}(z) \neq \pi_{\theta,x}(z)$ for any values of $\phi$. Note that this corresponds to Scenario 2 in Subsection A.2.

To initialise the gradient-ascent algorithm, we draw each component of the initial values $\phi_0$ of $\phi$ IID according to a standard normal distribution. We use both plain stochastic gradient-ascent with the gradients normalised to have unit $\mathbb{L}_1$-norm (Figures 1a, 2a) and ADAM (Kingma & Ba, 2015) with default parameter values (Figures 1b, The total number of iterations is $10,000$; in each case, the learning-rate parameters at the $i$th step are $i^{-1/2}$.

We also ran the algorithms in each of the above-mentioned scenarios with fixed values of $c_d$, e.g. as in Rainforth et al. (2018); Tucker et al. (2019). However, we omit the results as this did not significantly change the relative performance of the different algorithms. For the same reason, we omit results related to the optimisation of $A$ and $C$.

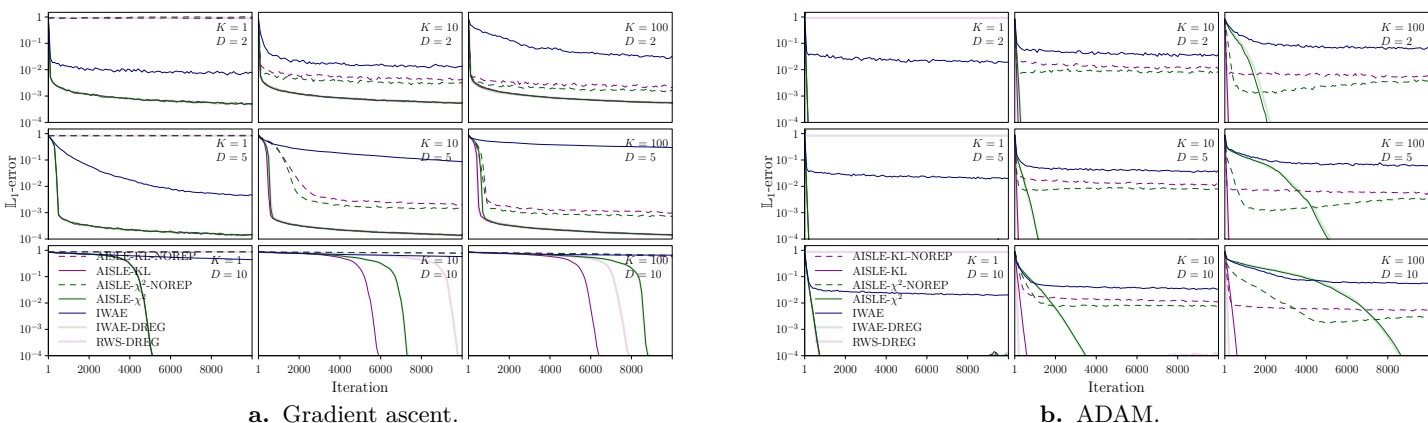

**Figure 1.** Average $\mathbb{L}_1$-error of the estimates of the parameters $b = b_{1:D}$ governing the mean of the Gaussian variational family. The average is taken over the $D$ components of $b$ and the figure displays the median error at each iteration over 100 independent runs of each algorithm, each using a different data set consisting of 25 observations sampled from the model. Note the logarithmic scaling on the second axis. Here, the covariance matrix $\Sigma = I$ *is diagonal.*

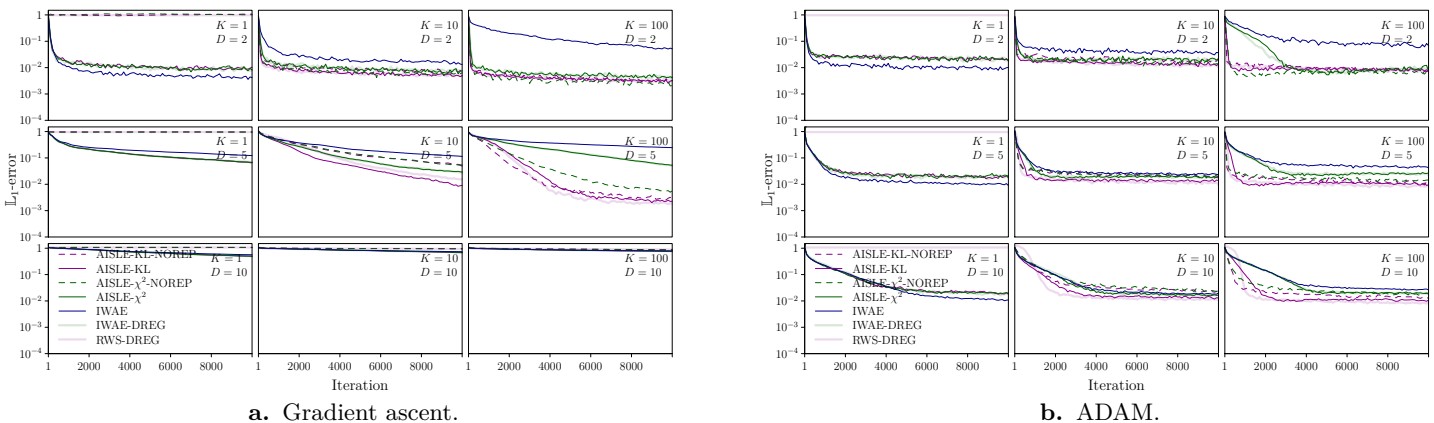

**Figure 2.** The same setting as in Figure 1 except that here, the covariance matrix $\Sigma = (0.95^{|d-e|+1})_{(d,e)\in\{1,\dots,D\}^2}$ *is not a diagonal matrix.* Again, note the logarithmic scaling on the second axis.

**Summary of results.**   Below, we outline what we believe to be the main takeaways from these simulation results for this particular model. However, further theoretical analysis is required to determine whether these hold in more general scenarios.

1. The 'score-function free' KL-divergence based AISLE algorithms typically performed somewhat better than their $\chi^2$-divergence based counterparts, i.e. AISLE-KL outperformed AISLE-$\chi^2$. We conjecture that this is due to the fact that the $\chi^2$-divergence based variants square the (self-normalised) importance weights which increases the variance of the $\phi$-gradients.

2. The performance of the $\phi$-gradients AISLE-KL-NOREP and AISLE-$\chi^2$-NOREP (which do not use any reparametrisation) typically benefited strongly from moderate (relative to the dimension of the latent variables) increases in the number of particles. In the scenario shown in Figure 2, for larger $K$, these gradients almost attained the performance of the 'score-function free' $\phi$-gradient AISLE-KL/IWAE-STL and outperformed AISLE-$\chi^2$/IWAE-DREG. We conjecture that this is due to the fact that in the scenario shown in Figure 2, the variational family does not include the target distribution, i.e. $q_\phi \neq \pi_\theta$ for any $\phi$, and as a result, the main advantage of the 'score-function free' gradients – i.e. the fact that they can potentially achieve zero variance – cannot be realised.

3. The standard IWAE $\phi$-gradient performed worse than the other methods in any of the scenarios considered (except in the trivial case $K = 1$ in which IWAE reduces to the VAE). Indeed, as expected, the performance of the standard IWAE $\phi$-gradient consistently worsened with increasing $K$. This can be attributed to the issue highlighted in Rainforth et al. (2018) (see Subsection 2.2), i.e. to the fact that the signal-to-noise ratio of this gradient vanishes as $\mathcal{O}(K^{-1/2})$ (as this gradient constitutes a self-normalised importance-sampling approximation of an integral which is equal to zero).

4. More surprisingly, the 'score-function free' $\phi$-gradients AISLE-KL/IWAE-STL, AISLE-$\chi^2$/IWAE-DREG did not necessarily improve with increasing $K$. Indeed, their performance sometimes became worse as can be seen most clearly in Figure 1. We note that this *cannot* be explained by the signal-to-noise ratio decay (which Rainforth et al. (2018) highlighted for the standard IWAE $\phi$-gradient) because the 'score-function free' $\phi$-gradients do not constitute self-normalised importance-sampling approximations of integrals which are equal to zero. Instead, we conjecture that as discussed in Remark 4 in the scenario shown in Figure 1, the $\mathcal{O}(K^{-1})$ self-normalisation bias of these gradients happens to be beneficial and outweighs the $\mathcal{O}(K^{-1/2})$ standard-deviation decrease obtained from increasing $K$.

5. The 'doubly-reparametrised' RWS-gradient RWS-DREG from Tucker et al. (2019) and given in (8) performed well for a moderate to large number of particles. Though it crucially requires $K > 1$ (note that for $K = 1$ this gradient is simply a vector of zeros).

