# OpenReview forum: "On importance-weighted autoencoders"
_ICLR.cc/2020/Conference — Reject_

### Official Review · AnonReviewer3 · 2019-10-22
**Official Blind Review #3**

**Rating:** 8

**Review:**

UPDATE: bumping up my score after the revisions

---


Nice connections but novelty and practical takeaways unclear

SUMMARY OF THE PAPER:

This paper views recent IWAE-based [1] methods (IWAE-STL [2], IWAE-DREG [3], RWS [4, 5]) for training generative models p and inference networks q under a common framework, AISLE.
This heavily relies on the "double-reparameterization" property by [2] and is restated in Lemma 1.
This framework makes it explicit that we're interested in
1) maximizing the (log) marginal likelihood wrt p parameters, and
2) minimizing some divergence between the posterior in the learned model to q.

In AISLE, IWAE-STL's q-gradient is viewed as a doubly-reparameterized self-normalized importance sampling (SNIS) estimate of KL(p || q).
This is in contrast to viewing it as a biased estimator of the IWAE's q-gradient.
This can potentially explain why it performs well when number of SNIS samples are increased.
It is also some evidence against the fact that having no unified objective is bad (because there isn't evidence of IWAE-STL diverging despite there being no unified objective).

IWAE-DREG's q-gradient is viewed as a doubly-reparameterized SNIS estimate of X-divergence(p || q) (up to multiplicative constant of the number of SNIS samples).
This is in contrast to viewing it as an unbiased estimator of the IWAE's q-gradient.

The view on RWS is unchanged: the q-gradient is a SNIS estimate of KL(p || q).

For me, the main contribution is viewing IWAE-STL and IWAE-DREG q-gradient estimators as biased gradients of an explicit divergence rather than of the IWAE objective.
I also found the observation that the signal-to-noise (SNR) decrease in IWAE's q-gradient can be proved by noting that it is a SNIS estimator of a zero vector nice.

STRUCTURE:
The article is well-written and easy to understand.

NOVELTY:
A different view on IWAE-STL and IWAE-DREG is interesting and novel (as mentioned above).
This means that IWAE-STL and IWAE-DREG are good not only because they reduce gradient variance (as previously understood) but also potentially because they directly target a divergence.
Viewing generalization of RWS as a main contribution (first bulletpoint of Section 1.2: "...we show that AISLE admits RWS as a special case.") is a bit of a stretch since this generalization is very straightforward from the way RWS is formulated.
The recommendation of using RWS-style algorithms over IWAE as given in the abstract ("we argue that directly optimising the proposal distribution in importance sampling as in the RWS algorithm is preferable to optimising IWAE-type multi-sample objectives) is also not novel since this is also advocated by [5] (section 3.2: "This makes RWS a preferable option to IWAE for learning inference networks because the phi updates in RWS directly target minimization of the expected KL divergences from the true to approximate posterior").
The recommendation as a method for non-reparameterisable latent variables at the end of section 1.2 ("as well as further algorithms which do not require reparameterisations") is also given in [5].
Are there different adaptive importance sampling algorithms that could be used within AISLE that would improve on IWAE-STL/IWAE-DREG/RWS?

EXPERIMENTS:
There are no experiments in the main paper.
However, experiments that would support/falsify the following points could be good:
- RWS and IWAE-STL don't suffer from non-unified objectives because IWAE-STL has non-unified objectives but doesn't diverge,
- [targeting direct divergence] is more useful than (or as useful as) [lower variance gradient estimators].

CONCLUSION:
While I really like the presentation and connections made in the paper, I'm not sure what the practical takeaways are (other than use IWAE-STL, IWAE-DREG, RWS over IWAE which is advocated by [2], [3], [5]).
I'm giving this a weak accept due to the former.
I'm willing to bump up my score if
- the paper is modified to more accurately reflect the contributions or
- there are experiments that provide additional support for the [targeting direct divergence] view in addition to [2, 3, 4, 5], or
- there is a new practical algorithm that the AISLE generalization would suggest that is better than IWAE-STL, IWAE-DREG, RWS in some respects.

[1] Importance Weighted Autoencoders. https://arxiv.org/abs/1509.00519
[2] Sticking the Landing: Simple, Lower-Variance Gradient Estimators for Variational Inference. https://arxiv.org/abs/1703.09194
[3] Doubly Reparameterized Gradient Estimators for Monte Carlo Objectives. https://arxiv.org/abs/1810.04152
[4] Reweighted Wake-Sleep. https://arxiv.org/abs/1406.2751
[5] Revisiting Reweighted Wake-Sleep for Models with Stochastic Control Flow. https://arxiv.org/abs/1805.10469
[6] Variational Inference via χ-Upper Bound Minimization. https://arxiv.org/abs/1611.00328
[7] Tighter Variational Bounds are Not Necessarily Better. https://arxiv.org/abs/1802.04537

**Experience Assessment:**

I have published one or two papers in this area.

**Review Assessment: Checking Correctness Of Derivations And Theory:**

I carefully checked the derivations and theory.

**Review Assessment: Checking Correctness Of Experiments:**

N/A

**Review Assessment: Thoroughness In Paper Reading:**

I read the paper thoroughly.

---

> ### Author Response · Authors · 2019-11-15
> **Thank you for providing such detailed and thoughtful feedback.**
>
> COMMENT:
> "I'm not sure what the practical takeaways are (other than use IWAE-STL, IWAE-DREG, RWS over IWAE which is advocated by [2], [3], [5])".
> REPLY:
> We agree that it is difficult to say which of the $\phi$-gradients one should use in practice (except that, as you say, the standard IWAE-gradient should be avoided due to its signal-to-noise ratio breakdown). Most likely, the answer to this is highly dependent on the specific application and choice of tuning parameters, e.g. on the number of particles. In our view, the main "practical" takeaway from our work is the following: If one is interested in the bias-reduction potential offered by IWAEs over plain VAEs then the adaptive importance-sampling framework appears to be a better starting point for designing new algorithms than the specific multi-sample objective used by IWAE. This is because the former retains all of the benefits of the latter without inheriting its drawbacks.
>
> COMMENT:
> "Viewing generalization of RWS as a main contribution (first bulletpoint of Section 1.2: "...we show that AISLE admits RWS as a special case.") is a bit of a stretch since this generalization is very straightforward from the way RWS is formulated."
> REPLY:
> An absolutely fair point. We have now removed this sentence to more accurately reflect the fact that the connection between RWS and AISLE is obvious and was not meant to be viewed as a "contribution" in itself.
>
> COMMENT:
> "The recommendation of using RWS-style algorithms over IWAE as given in the abstract ("we argue that directly optimising the proposal distribution in importance sampling as in the RWS algorithm is preferable to optimising IWAE-type multi-sample objectives) is also not novel since this is also advocated by [5] (section 3.2: "This makes RWS a preferable option to IWAE for learning inference networks because the phi updates in RWS directly target minimization of the expected KL divergences from the true to approximate posterior")."
> REPLY:
> Again a fair point. As you say, Le et al. (2019) [5] already demonstrated (based on extensive empirical studies) that RWS is often preferable to IWAE. Our work formalises this argument by showing that even the (heuristically) modified variants of IWAE (IWAE-DREG and IWAE-STL) which /do/ sometimes outperform "plain" RWS (e.g. in the numerical experiments in Tucker et al., 2019) can be derived in a principled manner from a suitably generalised version of RWS. Thus, our work is complementary to [5] (as a side node: [5] investigated only "plain" RWS -- as far as we know, our connection between the "score-function free" gradients AISLE-KL and AISLE-$\chi^2$ and RWS is novel). To address your comment, we have taken the sentence in question out of the abstract and are now explicitly mentioning in the introduction (on Page 2) that [5] already reached the conclusion that seeking to directly optimise the proposal distribution is often preferable to IWAE, especially because the former does not rely on reparametrisations.
>
> COMMENT:
> "The recommendation as a method for non-reparameterisable latent variables at the end of section 1.2 ("as well as further algorithms which do not require reparameterisations") is also given in [5]."
> REPLY:
> Also a fair point. We have now updated the introduction to reflect the fact that the advantage of RWS for non-reparametrisable models is already stressed in [5].
>
> COMMENT:
> "Are there different adaptive importance sampling algorithms that could be used within AISLE that would improve on IWAE-STL/IWAE-DREG/RWS?"
> REPLY:
> As mentioned in our reply to Reviewer 1, we have now also derived the $\phi$-gradients in the case of a class of $\alpha$-divergences. Furthermore, we have added a subsection discussing another special case which is obtained if we consider the "reverse" (i.e. "exclusive") KL-divergence $KL(q_\phi\|\pi_\theta)$. Here, the $\phi$-gradient reduces to the VAE-STL gradient proposed in Roeder et al. (2017, Equation 8) (more precisely, it reduces to a simple average over $K$ independent replicates of such VAE-STL estimators). As we discuss in Appendix A, optimising the "exclusive" KL-divergence can lead to faster convergence for $\phi$ than optimising the "inclusive" KL-divergence $KL(\pi_\theta\|q_\phi)$. However, care must be taken because minimising the "exclusive" KL-divergence does not can lead to poorly behaved or even ill-defined importance weights which can negatively affect learning of $\theta$ (whose gradient is an importance-sampling approximation which makes use of those weights).

---

### Official Review · AnonReviewer1 · 2019-10-23
**Official Blind Review #1**

**Rating:** 6

**Review:**

Summary: This paper presents a unifying framework through which much of the recent work on maximum likelihood learning in latent variable models (variational autoencoders) via multisample variational approaches and importance weighted approaches can be understood. It is a relatively clean framework that shows how many of the popular approaches can be described a distinct gradient-based approaches for a single underlying framework with two separate objectives for the generative model weights and the variational posterior weights.

Strengths:
- The framework is elegant and the derivations are simple.
-  It clarifies the connection between distinct algorithms and shows the consistency of ones that were previously poorly understood (IWAE-STL).
- It has the potential of generating new interesting algorithms.

Weaknesses:
- It is not clear to me why the first bullet of Remark 1 is so crucial. IWAE can be applied in discrete settings (see, e.g., Mnih & Rezende, 2016) and, as you show, reparameterizations can be applied in the adaptive importance sampling type algorithms to potential improve the variance.
- The paper currently struggles with some organizational issues. It reads as if the paper was written as a 15 page paper and split in half to satisfy the length requirements. In particular, I would recommend shortening the introduction, cutting out as much of Sec 2 as possible, and moving experiments into the main draft. Derivations are fine left in the Appendix.
- I appreciate that it wasn't the primary aim of the paper to introduce new algorithms, but I think it could strengthen the contribution to consider at least a few. Are there any other interesting divergences to consider for the proposal distribution objective?
- The experiments are quite lacking. In tandem with the above point (consider novel algorithms), the ICLR community might rightfully expect some experiments on large scale models.  I appreciate that there might not be much consistency in terms of which methods outperform others, but large scale experiments would at least present evidence of this point.

Citations:
Mnih & Rezende, 2016. https://arxiv.org/abs/1602.06725

**Experience Assessment:**

I have published in this field for several years.

**Review Assessment: Checking Correctness Of Derivations And Theory:**

I carefully checked the derivations and theory.

**Review Assessment: Checking Correctness Of Experiments:**

I assessed the sensibility of the experiments.

**Review Assessment: Thoroughness In Paper Reading:**

I read the paper at least twice and used my best judgement in assessing the paper.

---

> ### Author Response · Authors · 2019-11-15
> **Thank you for all your time and effort spent on reviewing our work.**
>
> COMMENT:
> "It is not clear to me why the first bullet of Remark 1 is so crucial. IWAE can be applied in discrete settings (see, e.g., Mnih & Rezende, 2016) and, as you show, reparameterizations can be applied in the adaptive importance sampling type algorithms to potential improve the variance."
> REPLY:
> What we were trying to say here is that whilst reparametrisations /can/ be used for adaptive importance-sampling approaches, they are not crucial (this is in contrast to IWAE in which the $\phi$-gradient incurs an additional high-variance term which is then removed through the reparametrisation). Following your suggestion, we have now clarified the first bullet point in Remark 1 to explain that control-variate approaches (Mnih & Rezende, 2016) can be used to reduce the variance of the IWAE $\phi$-gradient in scenarios in which reparametrisations are not available (though, Le et al., 2019 demonstrate that control-variate constructions or continuous relaxations are not always applicable or sufficient).
>
> COMMENT:
> "The paper currently struggles with some organizational issues. It reads as if the paper was written as a 15 page paper and split in half to satisfy the length requirements. In particular, I would recommend shortening the introduction, cutting out as much of Sec 2 as possible, and moving experiments into the main draft. Derivations are fine left in the Appendix."
> REPLY:
> Following your suggestion, we have now significantly reduced the length of the introduction and of Section 2. As our contribution is to establish links between existing algorithms in a formal manner, numerical results are not crucial and do not contribute much additional insight. Therefore, we opt to leave the numerical illustrations in Appendix B and instead use the available space to add discussions on other interesting divergences as you suggested in the comment quoted immediately below.
>
> COMMENT:
> "I appreciate that it wasn't the primary aim of the paper to introduce new algorithms, but I think it could strengthen the contribution to consider at least a few. Are there any other interesting divergences to consider for the proposal distribution objective?"
> REPLY:
> Following your suggestion, we have now also derived the $\phi$-gradients in the case of a class of $\alpha$-divergences. Furthermore, we have added a subsection discussing another special case which is obtained if we consider the "reverse" (i.e. "exclusive") KL-divergence $KL(q_\phi\|\pi_\theta)$. Here, the $\phi$-gradient reduces to the VAE-STL gradient proposed in Roeder et al. (2017, Equation 8) (more precisely, it reduces to a simple average over $K$ independent replicates of such VAE-STL estimators). As we discuss in Appendix A, optimising the "exclusive" KL-divergence can lead to faster convergence for $\phi$ than optimising the "inclusive" KL-divergence $KL(\pi_\theta\|q_\phi)$. However, care must be taken because minimising the "exclusive" KL-divergence can lead to poorly behaved or even ill-defined importance weights which can in turn negatively affect learning of $\theta$ (whose gradient is an importance-sampling approximation which makes use of those weights).
>
> COMMENT:
> "The experiments are quite lacking. In tandem with the above point (consider novel algorithms), the ICLR community might rightfully expect some experiments on large scale models. I appreciate that there might not be much consistency in terms of which methods outperform others, but large scale experiments would at least present evidence of this point."
> REPLY:
> We have now made it much more clear in the paper that some numerical illustrations are included in Appendix B and that extensive simulations for the main algorithms discussed in our work can be found in Le et al. (2019) and Tucker et al. (2019). Indeed, as pointed out by Reviewer 3, the argument -- supported by substantial numerical evidence -- that RWS's adaptive importance-sampling framework may be preferable to IWAE's multi-sample objective framework can already be found in Le et al. (2019). Our work simply provides a more formal framework for this argument for which numerical results are not crucial.

---

### Official Review · AnonReviewer2 · 2019-10-24
**Official Blind Review #2**

**Rating:** 3

**Review:**

Summary:
The authors review recent developments in gradient estimators for the IWAE bound and use them to develop a new theoretical justification for the Sticking the Landing (STL) estimator.

Unfortunately, the sole novelty in this paper is a new justification for the STL estimator. The presentation, while thorough, is not novel or particularly clear. There are no experiments. These factors combine to lead me to suggest a reject.

Specific points:
* The "AISLE framework" is simply used to point out that IWAE and RWS optimize KLs in different directions for the parameters of q. This is well-known in the literature and is discussed in several of the papers cited by the authors. A new framework is not needed to point this out.
* There seems to be an overall misunderstanding of the difficulties associated with multi-sample objectives. The difficulties with IWAE do not come because it is multi-sample, but because the KL direction optimized for the approximate posterior (q) is from q to p. Thus to compute gradients of the KL we must take gradients back through latent variables sampled from q. RWS avoids this by optimizing the other KL direction, and thus does not need to take gradients through the sampling operation. Many of the issues with IWAE mentioned in the paper also appear with the standard ELBO, which is a single-sample bound.
* Furthermore, IWAE does not necessarily require reparameterizations to deal with the high variance of its terms. Control variates can be used when latent variables are discrete, e.g. Mnih et. al 2016 "Variational inference for Monte Carlo objectives" and Tucker et al. 2017 "REBAR: Low-variance, unbiased gradient estimates for discrete latent variable models".
* RWS can still be thought of as a multi-sample objective in the sense that you use multiple samples of z to estimate the gradient for one data point x. The true difference is that the RWS gradient estimator is an asymptotically consistent estimator of the gradient of the marginal likelihood, while the IWAE gradient estimator is an unbiased estimator of the gradient of an objective (IWAE lower bound) that becomes the marginal likelihood in the limit of infinite samples.
* As such, your claim to "have shown that the adaptive-importance sampling paradigm of the reweighted wake-sleep is preferable to the multi-sample objective paradigm of importance weighted autoencoders" is far too strong, especially considering the fact that experimental evidence in Tucker et al. 2018 shows there are situations where either one is preferable. Additionally, the DReGS estimator avoids 2 of the 3 issues you present in remark 1.
* A smaller point: I found it hard to follow your derivations when compared with Tucker et al. 2018 because their identities use expectations over standard gaussian noise (epsilons) while expectations in your paper are all written with respect to q (e.g. the right-hand side of lemma 1). It would be helpful to go more in-depth about why that is.

To change my mind the authors would have to include experimental evaluation of some kind and demonstrate more novelty.

**Experience Assessment:**

I have published in this field for several years.

**Review Assessment: Checking Correctness Of Derivations And Theory:**

I carefully checked the derivations and theory.

**Review Assessment: Checking Correctness Of Experiments:**

I assessed the sensibility of the experiments.

**Review Assessment: Thoroughness In Paper Reading:**

I read the paper thoroughly.

---

> ### Author Response · Authors · 2019-11-15
> **Thank you for your extensive feedback. It has motivated us to clarify many of our arguments.**
>
> COMMENT:
> "Unfortunately, the sole novelty in this paper is a new justification for the STL estimator. The presentation, while thorough, is not novel or particularly clear."
> REPLY:
> We do not believe that this to be an accurate representation of our contributions. To our knowledge, the connections such as those formalised in Propositions 1 and 2 are indeed novel -- an assessment which does not appear to be contradicted by the other reviews. Taken together, these propositions make it clear that if one is interested in the bias-reduction potential offered by IWAEs over plain VAEs, then the adaptive importance-sampling framework is a more sensible starting point for designing new algorithms than the specific multi-sample objective used by IWAE. This is because the former retains all of the benefits of the latter without inheriting its drawbacks regarding the inference-network gradient (i.e. the fact that this gradient incurs high-variance term which must then be removed via reparametrisations and that it even then suffers from the signal-to-noise-ratio breakdown).
>
> Given the large number of papers published on IWAEs and extensions (the seminal paper: Burda et al. (2016), which introduced IWAEs currently has 450 Google Scholar citations), we believe that this point is of interest to researchers working on IWAEs and related methods.
>
> We would also be very happy to clarify the presentation if the reviewer could more concretely point out those places in which additional clarity is needed.
>
>
> COMMENT:
> "There are no experiments."
> REPLY:
> As we mention in our replies to the other reviewers, we have now made it much more clear in the paper that some numerical illustrations are included in Appendix B and that extensive simulations for the main algorithms discussed in our work can be found in Le et al. (2019) and Tucker et al. (2019). Indeed, as pointed out by Reviewer 3, the argument -- supported by substantial numerical evidence -- that RWS's adaptive importance-sampling framework may be superior to IWAE's multi-sample objective framework is already can already be found in Le et al. (2019). Our work simply provides a more formal framework for this argument for which numerical results are not crucial.
>
> COMMENT:
> "The "AISLE framework" is simply used to point out that IWAE and RWS optimize KLs in different directions for the parameters of q. This is well-known in the literature and is discussed in several of the papers cited by the authors. A new framework is not needed to point this out."
> REPLY:
> This is incorrect. What divergence (if any) these algorithms optimise depends on a number of factors including, crucially, the number of particles, $K$. In the trivial case $K=1$ IWAE reduces to a VAE and therefore does indeed minimise the "exclusive" KL-divergence $KL(q_\phi\|\pi_\theta)$. As discussed in more detail in our reply to the quoted comment immediately below, we can use the AISLE-framework to minimise the same divergence if we take the function $f$ in the $f$-divergence to be $f = - \log$.
>
> COMMENT:
> "There seems to be an overall misunderstanding of the difficulties associated with multi-sample objectives. The difficulties with IWAE do not come because it is multi-sample, but because the KL direction optimized for the approximate posterior (q) is from q to p. Thus to compute gradients of the KL we must take gradients back through latent variables sampled from q. RWS avoids this by optimizing the other KL direction, and thus does not need to take gradients through the sampling operation."
> REPLY:
> The difficulties with IWAE really /are/ a consequence of its specific multi-sample objective and not merely of the KL-direction. To see this, note that as we now discuss in Section 3.3.4 of the revised manuscript, we can use the AISLE-framework to optimise the "exclusive" KL-divergence $KL(q_\phi||\pi_\theta)$ by taking the function $f$ in the $f$-divergence to be $f = - \log$. If we do this, we obtain the analytical $\phi$-gradient
> $$
> -\nabla_\phi KL(q_\phi||\pi_\theta)
>  = q_\phi(\log w_\psi \nabla_\phi \log q_\phi)
>  = q_\phi(\blacktriangledown_\psi),
> $$
> where the last term assumes a reparametrisable proposal and where $\blacktriangledown_\psi$ is the "black triangle" defined on Page 5 of our manuscript. Either one of the last two expressions in the displayed equation can be trivially approximated by the vanilla Monte Carlo method with $K$ samples which immediately implies that the resulting $\phi$-gradient estimators have a signal-to-noise ratio which does not degenerate with $K$. Note also that the second gradient expression, $q_\phi(\blacktriangledown_\psi)$, in the above displayed equation indeed takes gradients back through latent variables sampled from $q_\phi$. This shows that neither the direction of the KL-divergence being optimised nor the reparametrisation is the cause of the IWAE $\phi$-gradient breakdown.

---

> > ### Author Response · Authors · 2019-11-15
> > **We apologise for the long reply**
> >
> >
> >
> > COMMENT:
> > "Many of the issues with IWAE mentioned in the paper also appear with the standard ELBO, which is a single-sample bound."
> > REPLY:
> > As we stated in the abstract/introduction, the goal of our work is to analyse algorithms which use $K$ samples to reduce the bias of the estimated parameters $\theta$ relative to the MLE (e.g. by obtaining a tighter lower evidence-lower bound than the standard VAE ELBO). We do not make any claims about whether or not seeking such bias-reductions is a sensible goal in the first place. All we are showing is that there is no "free lunch" in the sense that attempting to reduce this bias through the use of IWAE with $K>1$ samples
> >
> > (a) causes problems due to the nature of the IWAE multi-sample objective such as the signal-to-noise decay highlighted in Rainforth et al. (2018);
> >
> > (b) the best-case-scenario for IWAE which avoids the signal-to-noise ratio decay by using modified IWAE-gradients such as IWAE-STL or IWAE-DREG takes us back to the more classical adaptive importance-sampling setting (as proved in Propositions 1 and 2).
> >
> > COMMENT:
> > "Furthermore, IWAE does not necessarily require reparameterizations to deal with the high variance of its terms. Control variates can be used when latent variables are discrete, e.g. Mnih et. al 2016 "Variational inference for Monte Carlo objectives" and Tucker et al. 2017 "REBAR: Low-variance, unbiased gradient estimates for discrete latent variable models"."
> > REPLY:
> > Fair point. As mentioned in our reply to the other reviewers, we have now updated the manuscript to mention that control variates or continuous relaxations could alternatively be used to reduce the variance. Though as demonstrated in Le et al. (2019), these do not always lead to sufficient variance reductions or may not even be applicable.
> >
> > COMMENT:
> > "RWS can still be thought of as a multi-sample objective in the sense that you use multiple samples of z to estimate the gradient for one data point x."
> > REPLY:
> > Perhaps we should have more precisely stated what we mean by "multi-sample" objective. We have now made it clear throughout the manuscript that we are referring specifically to IWAE's multi-sample objective. Loosely speaking, the IWAE multi-sample yields $\phi$-gradients by /first/ approximating some idealised objective through importance-sampling and /then/ taking gradients of the resulting approxiation. This order of differentiation/approximation is responsible for the $\phi$-gradient problems. In contrast, RWS and its generalisations /first/ take gradients of some idealised objective and /then/ approximate the resulting expression by some Monte-Carlo method. Whilst the approximated RWS/AISLE $\phi$-gradient is then based on multiple (Monte Carlo) samples, we do not see how you could derive it by differentiating some Monte-Carlo approximation. The constructions given in Appendix 8.3 of Tucker et al. (2019) come closest to representing RWS/AISLE as multi-sample objective methods in the way you describe but this is only achieved by heuristically requiring that gradients are "stopped" w.r.t. certain random variables.
> >
> > COMMENT
> > "The true difference [between RWS and IWAE] is that the RWS gradient estimator is an asymptotically consistent estimator of the gradient of the marginal likelihood, while the IWAE gradient estimator is an unbiased estimator of the gradient of an objective (IWAE lower bound) that becomes the marginal likelihood in the limit of infinite samples."
> > REPLY:
> > We agree with this statement. Indeed, it highlights the two different philosophies/paradigmes at work here:
> > unbiased gradient of biased objective (IWAE) vs biased gradient of unbiased objective (adaptive importance sampling). However, our work shows that the differences between these two paradigms is not /only/ philosophical because the former paradigm introduces practical problems (e.g. the signal-to-noise decay) while the latter paradigm does not.

---

> > > ### Author Response · Authors · 2019-11-15
> > > **We apologise further for the long reply.**
> > >
> > > COMMENT:
> > > "As such, your claim to "have shown that the adaptive-importance sampling paradigm of the reweighted wake-sleep is preferable to the multi-sample objective paradigm of importance weighted autoencoders" is far too strong, especially considering the fact that experimental evidence in Tucker et al. 2018 shows there are situations where either one is preferable. Additionally, the DReGS estimator avoids 2 of the 3 issues you present in remark 1"
> > > REPLY:
> > > Indeed, the IWAE-DREG estimator avoids the signal-to-noise ratio breakdown. This is precisely the point of our paper and the whole reason why we introduced the AISLE-framework in the first place. Specifically, our work makes it clear that in all known cases in which the IWAE $\phi$-gradient can be modified to avoid the breakdown, the resulting estimators (AISLE-KL/IWAE-STL and AISLE-$\chi^2$/IWAE-DREG) can be much more straightforwardly derived as a special case of the adaptive importance-sampling framework (without having to heuristically drop terms such as needed to obtain IWAE-STL from IWAE). Furthermore, the IWAE $\phi$-gradient is consistently outperformed by AISLE-KL/IWAE-STL and AISLE-$\chi^2$/IWAE-DREG in Tucker et al. (2019), i.e. by algorithms that can be derived without the need for the IWAE-paradigm (it is true that the "plain" RWS algorithm breaks down in one of their examples but we do not claim that this particular instance of AISLE is preferable).
> > >
> > > COMMENT:
> > > "A smaller point: I found it hard to follow your derivations when compared with Tucker et al. 2018 because their identities use expectations over standard gaussian noise (epsilons) while expectations in your paper are all written with respect to q (e.g. the right-hand side of lemma 1). It would be helpful to go more in-depth about why that is."
> > > REPLY:
> > > We appreciate your concern. We believe that our notation and style of presentation is sensible because it is concise while also being general enough to accommodate arbitrary reparametrisations, i.e. these do not need to be based around standard Gaussian noise.

---

### Decision · Program_Chairs · 2019-12-19

**Decision:**

Reject

**Comment:**

The authors argue that directly optimizing the IS proposal distribution as in RWS is preferable to optimizing the IWAE multi-sample objective. They formalize this with an adaptive IS framework, AISLE, that generalizes RWS, IWAE-STL and IWAE-DREG.

Generally reviewers found the paper to be well-written and the connections drawn in this paper interesting. However, all reviewers raised concerns about the lack of experiments (Reviewer 3 suggested several experiments that could be done to clarify remaining questions) and practical takeaways.

The authors responded by explaining that "the main "practical" takeaway from our work is the following: If one is interested in the bias-reduction potential offered by IWAEs over plain VAEs then the adaptive importance-sampling framework appears to be a better starting point for designing new algorithms than the specific multi-sample objective used by IWAE. This is because the former retains all of the benefits of the latter without inheriting its drawbacks." I did not find this argument convincing as a primary advantage of variational approaches over WS is that the variational approach optimizes a unified objective. At least in principle, this is a serious drawback of the WS approaches. Experiments and/or a discussion of this is warranted.

This paper is borderline, and unfortunately, due to the high number of quality submissions this year, I have to recommend rejection at this point.